# Structure-Based Design, Virtual Screening, and Discovery of Novel Patulin Derivatives as Biogenic Photosystem II Inhibiting Herbicides

**DOI:** 10.3390/plants13121710

**Published:** 2024-06-20

**Authors:** He Wang, Jing Zhang, Yu Ji, Yanjing Guo, Qing Liu, Yuan Chang, Sheng Qiang, Shiguo Chen

**Affiliations:** Weed Research Laboratory, Nanjing Agricultural University, Nanjing 210095, China; 2018216001@njau.edu.cn (H.W.); 2022116023@stu.njau.edu.cn (J.Z.); 2023816112@stu.njau.edu.cn (Y.J.); 2021216003@stu.njau.edu.cn (Y.G.); liuqing@stu.njau.edu.cn (Q.L.); 2023116006@stu.njau.edu.cn (Y.C.); wrl@njau.edu.cn (S.Q.)

**Keywords:** natural product, photosynthetic inhibitor, D1 protein, docking, homology modeling

## Abstract

Computer-aided design usually gives inspirations and has become a vital strategy to develop novel pesticides through reconstructing natural lead compounds. Patulin, an unsaturated heterocyclic lactone mycotoxin, is a new natural PSII inhibitor and shows significant herbicidal activity to various weeds. However, some evidence, especially the health concern, prevents it from developing as a bioherbicide. In this work, molecular docking and toxicity risk prediction are combined to construct interaction models between the ligand and acceptor, and design and screen novel derivatives. Based on the analysis of a constructed patulin–*Arabidopsis* D1 protein docking model, in total, 81 derivatives are designed and ranked according to quantitative estimates of drug-likeness (QED) values and free energies. Among the newly designed derivatives, forty-five derivatives with better affinities than patulin are screened to further evaluate their toxicology. Finally, it is indicated that four patulin derivatives, D3, D6, D34, and D67, with higher binding affinity but lower toxicity than patulin have a great potential to develop as new herbicides with improved potency.

## 1. Introduction

Natural products are a key resource for developing new drugs and pesticides [1,2]. However, some active ingredients face the challenge of unstable activity, toxicity to non-targeting organisms, or environmental concerns [3]. Structure-based methods such as docking and molecular dynamics play a crucial role in computer-aided new drug design, enhancing our understanding of how small molecules bind to the protein targets. First of all, the strategy of such drug design is generally docking an interesting leading active ingredient or its analogues into a recognized target protein [4]. The development of structural biology, in that more and more experimentally determined structures of high-resolution protein crystal structures are analyzed via X-ray crystallography, nuclear magnetic resonance (NMR) spectroscopy, or cryo-electron microscopy (cryo-EM) methods and available in databases such as the Protein Data Bank (PDB) (http://www.wwpdb.org) or AlphaFold 2 (https://alphafold.com/), which are filled with more than 200 million structures, facilitates structure-based ligand design studies [4,5,6].

Photosystem II (PSII) is a multi-subunit pigment–protein complex embedded in the thylakoid membranes of plants, cyanobacteria, and algae, which is considered as the crucial photosynthetic component, catalyzing light-driven water-splitting and oxygen-evolving reactions, thereby converting light energy into electrochemical potential energy and generating molecular oxygen [7,8,9]. In PSII, Q_A_ and Q_B_ are the primary and secondary plastoquinone receptors, respectively. Many classical PSII inhibiting herbicides bind at the Q_B_ site of the D1 protein and interfere with the photosynthetic electron transport in competition with the native electron acceptor plastoquinone, and thereby block the transfer of electrons from Q_A_ to Q_B_. The Q_B_ site is located in the connecting loop between the fourth and the fifth transmembrane helices between the amino acid residue from Phe211 to Leu275 in plants. The classical urea/triazine herbicides including atrazine and DCMU with the common functional group N-C=R (R refers to an atom of oxygen or nitrogen) can form a hydrogen bond with D1-Ser264. The phenol-type herbicides with an aromatic hydroxyl group such as dinoseb and ioxynil form a hydrogen bond towards D1-His215 [10,11,12,13].

The mycotoxin patulin, an electrophilic unsaturated heterocyclic lactone (4-hydroxy-4H-furo[3,2-c]pyran-2(6H)-one), is widely present in final products, especially in apples and apple-based products, and predominantly produces by several fungal species involving *Aspergillus*, *Byssochlamys*, and *Penicillium* during storage [14,15]. Patulin can induce the generation of cytotoxin-involving reactive oxygen species (ROS), cytochrome c release in mitochondria, the uptake of cytosolic Ca^2+^, caspase-3 activation, cell cycle arrest, and cell apoptosis [16,17]. Furthermore, patulin causes the destruction of DNA damage, chromosome abnormalities, and the appearance of micronuclei in mammalian cells [18,19,20,21]. Early in the past century, it was indicated that patulin exhibited the potential not only in inhibiting seed germination but also seedling growth of wheat, indicating an interesting herbicidal activity [22]. Our previous research indicates that patulin is a novel natural PSII inhibitor and shows significant herbicidal activity to various weeds, especially the invasive weed *Ageratina adenophora*. The unsaturated lactone containing the C=O group at the 2-position is the essential part that bounds to the Q_B_ site through forming hydrogen bonds to D1-His252 [23]. But many cell line-based and animal model-based findings are evidence that patulin has the following properties: genotoxicity, embryotoxicity, cytotoxicity, neurotoxicity, immunotoxicity, carcinogenicity, and teratogenicity. These prevent it from developing into a bioherbicide due to the health concern [24]. Nevertheless, this natural product is still an interesting leading compound in discovering new active ingredients with high activity in weed control.

In this work, we focused on discovering novel highly bioactive and low-toxicity patulin derivatives by the strategy of structure-based ligand design. Firstly, eighty-one patulin derivatives, which are different introduced groups at 1-, C3, C4, 5-, C6, or C7 positions of patulin, were designed based on the simulated model of patulin and *Arabidopsis* D1 protein by computer-aided design. Subsequently, each designed derivative exhibiting higher binding affinity than patulin was screened through docking it into the D1 protein and its binding interaction energy calculation. The toxicity risks of forty-five patulin derivatives, which show a higher quantitative estimate of drug-likeness (QED) and binding affinity than patulin, were predicted by Discovery Studio software version 2016. Finally, four patulin derivatives involving D3, D6, D34, and D67 compounds with higher binding affinity but lower toxicity than patulin were used in the Q_B_ binding site of *Arabidopsis* D1 protein modeling to further analyze the detail feature of molecular interaction deeply. Obviously, structure-based ligand design is an efficient and highly reliable strategy to discover novel derivatives with high herbicidal activity through the core leading scaffolds of mycotoxin patulin. Figure 1 displays a methodology with a brief description of this study.

## 2. Results and Discussion

### 2.1. Model-Based Ligand Design and Molecular Docking of Patulin and Its Derivatives

Previous studies indicated that patulin binds to D1 protein, which blocks PSII electron transport beyond Q_A_. The simulated molecular modeling of patulin docking to D1 protein of *A. adenophora* suggests that D1-His252 plays an important role in the docking model of patulin and D1 protein [17] (Figure 2A). In this study, the crystal structure information of the model plant *Arabidopsis* D1 protein (PDB: 5MDX) was chosen to model the position of patulin in the Q_B_ site by Discovery Studio (Figure 2B–D). In this model, the residues including Phe211, Met214, Leu218, His252, Phe255, Ser264, Phe265, Leu271, Phe274, and Leu275 in the D1 protein are identified as responsible for patulin binding. The residue D1-His252 contributes to binding to patulin by forming a hydrogen bond with the O2 carbonyl oxygen atom of patulin. This result is congruent with early evidence from modeling of patulin binding to the D1 protein of *A. adenophora* [17].

The technique of molecular docking is extensively employed for the prediction of ligand orientation and conformation with the active site of the target protein, as well as for virtual screening based on the analysis of ranking scores or binding interaction energy of the docking model of the ligand–receptor complex [25]. On the basis of this method, two novel TeA analogues possessing higher herbicidal activity were discovered [26]. Yang et al. elucidated the mode of action of mycotoxin citrinin, and furtherly screened five new citrinin derivatives by introducing different substituents at the 9-position with high herbicidal potency compared with lead compound citrinin based on a computer-aided design strategy [27]. To find novel patulin derivatives with improved potency, QED values and binding interaction energy of 81 novel compounds were designed based on the modification of substituents in 1-, C3, C4, 5-, C6, or C7 positions of patulin binding to the Q_B_ site and are displayed (Table 1, Table 2, Table 3, Table 4, Table 5 and Table 6). Each compound with negative binding energy means that a favorable thermodynamic binding occurred between the ligand and D1 protein.

Compounds D1–D9 in Table 1 were introduced involving different substituents at the 1-position of patulin. Compounds with a sulfur (D1), methylamino (D3), alkyl side chains (D4–D6), alkyl chloride (D7) or hydroxy group (D8) exhibit lower interaction energies than patulin in the D1 protein. Compounds D2 and D9 with an amidogen or aminomethyl group at the 1-position of patulin show lower QED and higher interaction energy than patulin, indicating that these two compounds may have lower herbicidal activity than patulin.

Compounds D10–D19 in Table 2 focused on introducing different substituents at the C3 position of patulin. Compounds with a methyl (D10), methylthio (D12), ester (D16), acylamide (D18), and chlorine group (D19) exhibit higher QED and lower interaction energies than patulin. Compounds D11, D13–D15, and D17 with a sulfhydryl, ether, aldehyde, carboxy, or amino group at the C3 position of patulin have lower QED and higher interaction energy than patulin.

Compounds D20–D40 in Table 3 focused on introducing different substituents at the C4 position of patulin. Compounds D26, D29–D37, and D39 with an acyl chloride, alkyl side chain, chlorine or methylthio group exhibit higher QED and lower interaction energies than patulin, suggesting that these compounds show higher inhibition to PSII compared with patulin. On the other hand, D20–D25, D36, D38, and D40 with an ether, hydroxy, aldehyde, carboxy, chloride, sulfhydryl, or amino group introduced to the C4 position of patulin show a lower value of QED and higher interaction energy than patulin.

The QED and binding interaction energy of different substituents introduced at the 5-position of patulin were displayed in Table 4. Compounds D41 and D43–D46 with hydrophobic groups including sulfur, methylamino, or alkyl side chain groups exhibit a higher QED value and lower interaction energies than patulin. D42, D44, and D47–D49 with a hydrophilic group, such as an amino, chloride, or hydroxy group introduced to the 5-position of patulin, show a lower value of QED and higher interaction energy than patulin.

The QED and binding interaction energy of different substituents introduced at the C6 position of patulin were shown in Table 5. Compounds with an alkyl side chain (D50–D53) and a chloride (D54), ether (D56 and D57), carbonyl (D59), and methylthio (D65 and D66) group exhibit a higher QED value and lower interaction energies than patulin. D55, D58, D60–D63, and D68 with a hydroxy, aldehyde, carboxy, sulfhydryl, ester, or amino group at the C6 position of patulin have a lower value of QED and higher interaction energy relative to patulin.

Compounds D69–D81 in Table 6 focused on the modification of different substituents at the C7 position. Compounds with an alkyl side chain (D69 and D70) and chloromethyl (D71), ether (D73), carbonyl (D75), ester (D77), or methylthio (D79) group exhibit a higher QED value and lower interaction energies than patulin. D72, D74, D76, D78, and D80–D81 with a hydroxy, aldehyde, carboxy, sulfhydryl, amino, or chloride group at the C7 position of patulin show a lower value of QED and higher interaction energy with comparison to patulin.

Among these 81 derivatives, forty-five derivatives as possible candidates show a higher QED value and binding affinity than leading compound patulin. The QED value, which integrated eight key physicochemical properties of a drug candidate, provides an efficient approach to assessing drug-likeness and chemical attractiveness [28]. A greater QED value means a higher drug-likeness, representing a compound as more bioactive [29]. On the basis of the analysis, we found that the substituents including an alkyl side chain and methylthio group introduced to the patulin can increase the QED value of the new compound, such as compounds D4–D6 (1-), D10 (C3), D12 (C3), D29–D35 (C4), D39 (C4), D44–D46 (5-), D50–D53 (C6), D65–D67 (C6), D69 (C7), D70 (C7), and D79 (C7). Methylamino and acylamide groups introduced to the patulin at the 1-, C3, or 5-position, such as compounds D3, D18, and D43, can increase the QED value. The QED value of the compounds will be increased while the sulfur is introduced in the patulin at the 1- or 5-position, e.g., compounds D1 and D41. Chloride and alkyl chloride groups are introduced in the patulin at the C3, C4 (CH_2_Cl), or C6 (CH_2_Cl) position involving compounds D19, D37, and D54 and can increase the QED value. The introduction of an ether group at the C6 or C7 position can also increase the QED value including compound D56, D57, and D73. The QED values of compounds are increased when an ester group is introduced at the C3 or C7 position. However, the introducing of hydrophilic groups involving hydroxy, aldehyde, carboxy, sulfhydryl, or amino leads to a decrease in the QED value and the binding affinity. We expect that these introduced groups, which can increase the QED value of new compounds, may donate electrons by resonance to the C=O bond at the 2-position, or to the double bonds at C7 up to the carboxyl bond, resulting in increasing charge transfer between the proton donor and acceptor molecules and reinforcing hydrogen bond formation with D1-His252 [30].

### 2.2. Prediction of Toxicity of Patulin’s Derivatives by Computational Analysis

However, to ensure safe use of herbicides, strategies to assess the possible risks need to be developed. The characterization of the risk for non-target organisms is essential. The AAL toxin produced by *A. alternata* f. sp. *Lycopersici* has been known as an excellent herbicidal compound for a long time. However, its toxicity to cultured mammalian cells prevents it from developing as a bioherbicide. Recently, a few AAL toxin derivatives were designed and synthesized. Based on a series of experiments, it was indicated that one of them exhibited higher significant phytotoxicity but lower mammalian toxicity than the lead compound, which exhibited a potential for being developed as a safe and effective herbicide [31]. So, some toxicity risk parameters including the median lethal dose (LD_50_) of rat oral and rat inhalational LD_50_, skin irritancy, skin sensitization, ocular irritancy, or Ames mutagenicity of patulin and its forty-five derivatives were also predicted through the tools in Discovery studio. To validate the reliability of prediction, the values of rat oral LD_50_ and Ames mutagenicity were successfully obtained from PubChem Database (https://pubchem.ncbi.nlm.nih.gov/source/hsdb/3522#section=Animal-Toxicity-Studies, accessed on 15 April, 2024) and previous studies. As shown in Table 7, patulin has the rat oral LD_50_ of 55 mg/kg and no mutagenicity. Based on the Data Requirements for Pesticide Registration, the rating of rat oral LD_50_ of patulin is a moderate toxicity level (50–500 mg/kg). Here, the predicted rat oral LD_50_ of patulin is 139 mg/kg; moreover, it also does not have a mutagenic effect. Undoubtedly, our result is consistent with the existing data, showing high reliability and accuracy. On the basis of these predicted data, there are 39 compounds exhibiting a higher rat oral value than patulin, indicating lower oral toxicity. Especially, compounds D1, D3, D4, D6, D8, and D34 have an oral LD_50_ of over 500 mg/kg, which are of low toxicity. Among the predicting data of rat inhalational LD_50_, only the value of compound D54 is lower than 2000 mg/m^3^, which is of moderate toxicity (200–2000 mg/m^3^) in rats. The inhalational LD_50_ values of other patulin derivatives are higher than 2000 mg/m^3^, which is of low toxicity (2000–5000 mg/m^3^) or mild toxicity (>5000 mg/m^3^) in rats. Based on the data of skin irritancy, D31 shows severe irritancy to skin, and compounds D1, D32, D33, and D35 show moderate irritancy to skin. Other patulin derivatives are of mild irritancy to skin. According to the predicting data of skin sensitization, only D1 shows moderate sensitization to skin. The other 44 patulin’s derivatives are of weak or no sensitization to skin. More than 73% of patulin derivatives in Table 7 exhibit severe ocular irritancy. Compounds D16, D19, D56, D57, D59, and D77 show moderate ocular irritancy. Seven patulin derivatives involving D3, D6, D18, D34, D35, D39, and D67 exhibit mild ocular irritancy. The data about Ames mutagenicity indicate that all derivatives of patulin are non-mutagen.

All things considered, four derivatives, D3, D6, D34, and D67, among forty-five candidates were screened out as real candidates since they possess a high QED value and binding affinity as well as reasonable low animal toxicity.

### 2.3. Verification of the Molecular Models of Real Candidates’ Binding to D1 Protein

Furthermore, the molecular models of compound D3, D6, D34, and D67, which bind to the Q_B_ site of *Arabidopsis* D1 protein (5MDX), were constructed in order to investigate the properties of these four real candidates further. The binding energies of each docked pose are −26.871 kal·mol^−1^ (D3), −34.373 kal·mol^−1^ (D6), −33.048 kal·mol^−1^ (D34), and −35.601 kal·mol^−1^ (D67), which were displayed in Table 8. There is a single hydrogen bond between the residue His252 of D1 protein and the O2 carbonyl oxygen atom of these four compounds, while several kinds of hydrophobic interactions also occur with other residues of D1 protein (Figure 3). The modeled distance of their hydrogen bonds is 2.27 Å (D3), 2.13 Å (D6), 2.18 Å (D34), and 2.06 Å (D67), respectively, which is less than that of D0 (patulin), which is 2.38 Å. A shorter hydrogen bond distance may reinforce these candidates’ binding stability in the D1 protein. The molecular physicochemical properties of a compound play a crucial role in the determination of its binding affinity for the target protein [32]. Our results show a positive relationship between QED values and the binding affinity, and a negative relationship between QED values and the distance of a hydrogen bond. It was concluded that compounds D3, D6, D34, and D67 with a higher QED value improved the binding affinity of D1 protein due to having shorter hydrogen bond distances compared with lead compound patulin. As a consequence, these four derivatives, D3, D6, D34, and D67, have potential to be developed into biogenic photosynthetic herbicides.

## 3. Materials and Methods

### 3.1. Software

Molecular modeling studies are carried out through Discovery Studio (version 2016, BIOVIA, San Diego, CA, USA). The structures of various candidate ligands are created using ChemDraw 18.0 (Cambridge Soft, Cambridge, MA, USA). The ligand structures are optimized using MM2 energy minimizations by Chem3D Pro 14.0 (Cambridge Soft, USA).

### 3.2. The Preparation of Ligands

After energy minimization and the transformation of each ligand into the Discovery Studio program, the QED value was calculated according to our previous research [26].

### 3.3. The Preparation of Protein Data

Due to the available data of the crystal structure of *Arabidopsis* D1 protein, its high-resolution crystal structures were obtained from the database of Protein Data Bank (https://rcsb.org, PDB code: 5MDX, resolution: 5.30 Å), which was used as a template for the Dl protein according to our previous research [26]. Its dimeric structure was simplified to a monomer and optimized using Discovery Studio’s tools.

### 3.4. Interaction Model of Patulin Binding to D1 Protein of A. thaliana

The automation of the preliminary docking process was performed through Discovery Studio according to our previous research [26]. After the docking was finished, it was indicated that six residues, Met214, His252, Phe255, Ser264, Phe265, and Leu271, in the D1 protein might provide bonds to patulin molecules. Thus, the binding site for the accuracy of docking of patulin and these six residues is focused on through the CDocker tool of Discovery Studio. These six amino acid residues create a binding site sphere with a radius of 12.34 Å and coordinate axes of x = 301.43 Å, y = 242.62 Å, and z = 239.58 Å. Furthermore, the optimal binding pose, with the lowest interaction energy, was selected as the favorable binding configuration between patulin and the Q_B_ binding site of the *A. thaliana* D1 protein. The 2D and 3D diagrams were opted for to illustrate the ligand binding site atoms in the molecular interaction model.

### 3.5. Model-Based Ligand Design and Molecular Docking of Patulin Derivatives

On the basis of our previous studies of the patulin–D1 protein binding model, it was indicated that the O2 carbonyl oxygen atom is a core scaffold for patulin’s biological activity, which forms a major hydrogen bond between the residue D1-His252 [23]. Consequently, the substituents at 1-, C3, C4, 5-, C6, or C7 positions of patulin were modified to find possible novel derivatives with high potency based on the molecular interaction modeling of patulin–*A. thaliana* D1 protein. The 81 patulin’s derivatives designed and energetically minimized through ChemDraw 18.0 and Chem3D Pro 14.0, respectively, were divided into six types on the basis of the substituents modifying in 1-, C3, C4, 5-, C6, or C7 positions. Then, the molecular properties of each ligand were evaluated by Discovery Studio. Subsequently, we hypothesized that these designed derivatives show a similar binding manner with patulin at the Q_B_ side of *A. thaliana* D1 protein. Notably, the binding site, which forms by the residues including D1-Met214, D1-His252, D1-Phe255, D1-Ser264, D1-Phe265, and D1-Leu271, was delineated for these derivatives. The method of the molecular docking of these 81 derivatives is similar to that described in the “*3.4 Interaction Model of Patulin Binding to D1 Protein of A. thaliana*” section. The interaction energy of each ligand at the Q_B_ binding site of *A. thaliana* D1 protein was calculated and is presented in Table 1.

### 3.6. Prediction of Toxicity of Patulin Derivatives by Computational Analysis

Due to the current requests for environmental protection and safe food production, the most important pesticide properties used in pesticide registration refer to physical and chemical characteristics, toxicology and metabolism, environmental fate and behavior, and ecotoxicology (https://www.fao.org/pesticide-registration-toolkit/information-sources/pesticide-properties/en/, accessed on 22 April 2024). The toxicity of pesticides is closely related to their structure. Structure–activity relationships (SARs) have been widely used in Europe and the United States to predict toxicity by a computer [33]. To further confirm the toxicity of patulin derivatives, patulin and its 45 derivatives, which exhibit higher QED and binding affinity values than patulin, were selected to systematically discuss the structure–toxicity relationship. Six parameters involving the rat oral and rat inhalational lethal dose (LD_50_), skin irritancy, skin sensitization, ocular irritancy, and Ames mutagenicity of patulin and its 45 derivatives were calculated using Discovery Studio’s tools. To validate the reliability of the toxicity of patulin and its 45 derivatives, the values of rat oral LD_50_ and Ames mutagenicity were successfully obtained from PubChem Database (https://pubchem.ncbi.nlm.nih.gov/source/hsdb/3522#section=Animal-Toxicity-Studies, accessed on 22 April 2024) and previous studies [34]. The toxicity classification of patulin and its 45 derivatives referred to the Data Requirements for Pesticide Registration (http://www.chinapesticide.org.cn/zgnyxxw/zwb/detail/17809, accessed on 22 April 2024).

## 4. Conclusions

So, patulin can indeed be used as a lead compound to discover novel derivatives with high potency. In this work, four novel compounds with substituents of methylamino (D3) or propyl (D6) at the 1-position, sec-butyl at the 4-position (D34), or methylthio at the 9-position (D67) exhibited higher predicted binding affinity to the Q_B_ site than patulin. However, further studies are required to synthesize these compounds and to evaluate their PSII inhibitory capacity, herbicidal activity, and animal toxicology for possible commercial development.

## Figures and Tables

**Figure 1 plants-13-01710-f001:**
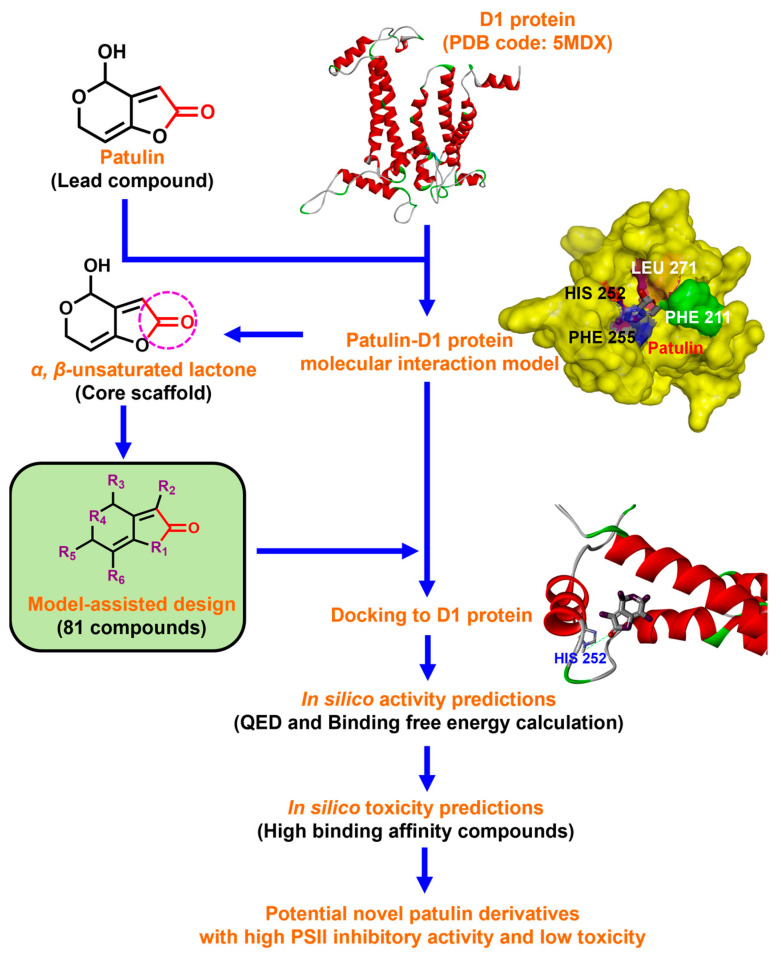
Flow chart illustrating the structure-based ligand design and discovery of novel patulin derivatives with high herbicidal activity.

**Figure 2 plants-13-01710-f002:**
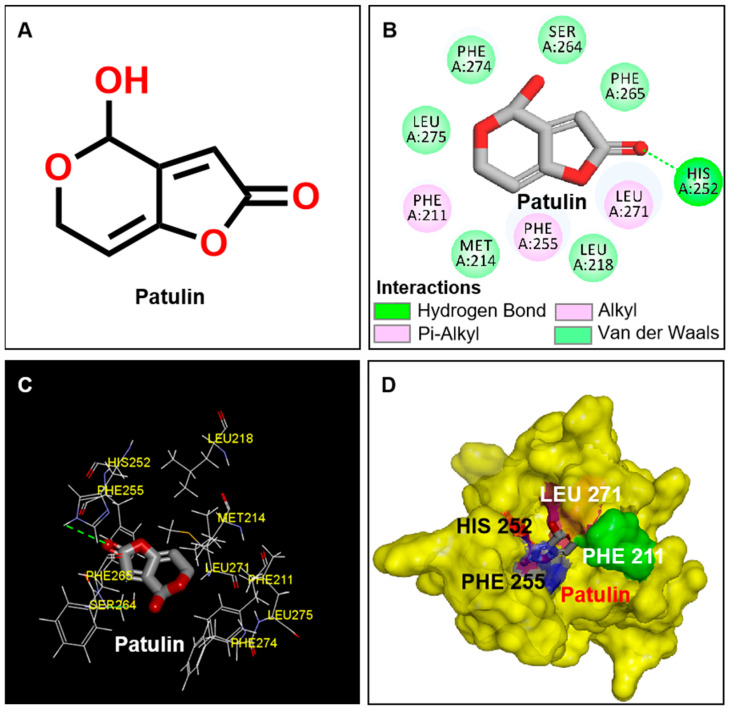
Simulated modeling of patulin binding to the D1 protein of *Arabidopsis*. (**A**) The chemical structure of patulin. (**B**) Hydrogen bonding interactions of patulin binding to the D1 protein. (**C**) The stereo view of the patulin binding environment of the D1 protein, in which carbon, oxygen, nitrogen, and hydrogen atoms are displayed in gray, red, blue, and white, respectively. The green dashed lines represent the possible hydrogen bonds. (**D**) The surface representation of the Q_B_ binding site with bound patulin.

**Figure 3 plants-13-01710-f003:**
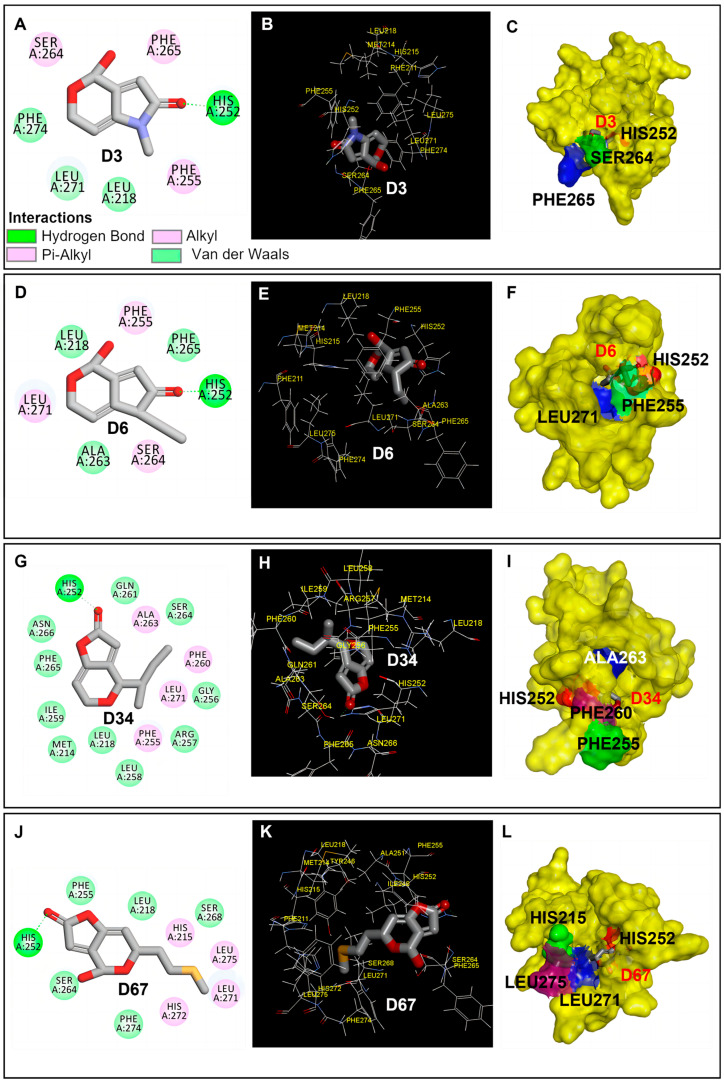
Binding interactions of patulin derivatives at the Q_B_ binding site of D1 protein of *Arabidopsis*. An illustration of the binding mode of compounds D3 (**A**), D6 (**D**), D34 (**G**), and D67 (**J**) binding to the D1 protein, respectively. Key interaction types are represented in the color code. The stereo view of compound D3 (**B**), D6 (**E**), D34 (**H**), and D67 (**K**) binding environments at the Q_B_ binding site. The surface representation of the Q_B_ binding site with compounds D3 (**C**), D6 (**F**), D34 (**I**), and D67 (**L**), respectively.

**Table 1 plants-13-01710-t001:** Structures and QED of new patulin’s derivatives, which have different R group at 1-position, and the binding affinity between the compounds and *Arabidopsis* D1 protein (5MDX). Seven derivatives (D1, D3, D4, D5, D6, D7, and D8) show higher QED and binding affinity than patulin.

Chemical Structure	Compound No.	Substituent (R-)	QED	Interaction Energy (kal·mol^−1^)
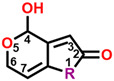	D0 (patulin)	O	0.488	−25.383
	D1	S	0.566	−29.405
	D2	NH	0.473	−22.093
	D3	N-CH_3_	0.518	−26.871
	D4	CH_2_	0.530	−27.825
	D5	CHCH_3_	0.567	−29.641
	D6	CHCH_2_CH_3_	0.651	−34.373
	D7	CHCl	0.561	−28.940
	D8	CH-OH	0.503	−26.086
	D9	CH-NH_2_	0.487	−25.225

**Table 2 plants-13-01710-t002:** Structures and QED of new patulin’s derivatives, which have different R group at C3 position, and the binding affinity between each compound and *Arabidopsis* D1 protein (5MDX). Five derivatives (D10, D12, D16, D18, and D19) show higher QED and binding affinity than patulin.

Chemical Structure	Compound No.	Substituent (R-)	QED	Interaction Energy (kal·mol^−1^)
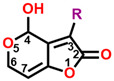	D0 (patulin)	H	0.488	−25.383
	D10	CH_3_	0.523	−27.349
	D11	SH	0.403	−17.148
	D12	S-CH_3_	0.601	−31.482
	D13	CO-CH_3_	0.448	−19.228
	D14	CHO	0.330	−15.742
	D15	COOH	0.428	−18.243
	D16	CO-OCH_3_	0.492	−25.682
	D17	NH_2_	0.432	−18.782
	D18	NH-COCH_3_	0.518	−26.945
	D19	Cl	0.548	−28.307

**Table 3 plants-13-01710-t003:** Structures and QED of new patulin’s derivatives, which have different R group at C4 position, and the binding affinity between each compound and *Arabidopsis* D1 protein (5MDX). Ten derivatives (D26, D29, D30, D31, D32, D33, D34, D35, D37, D39) show higher QED and binding affinity than patulin.

Chemical Structure	Compound No.	Substituent (R-)	QED	Interaction Energy (kal·mol^−1^)
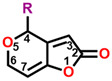	D0 (patulin)	OH	0.488	−25.383
	D20	O-CH_3_	0.470	−21.481
	D21	O-CH_2_CH_3_	0.481	−24.653
	D22	O-CH_2_CH_2_CH_3_	0.482	−24.804
	D23	OH	0.487	−25.092
	D24	CHO	0.396	−16.454
	D25	CO-CH_3_	0.422	−17.838
	D26	CO-Cl	0.501	−26.005
	D27	COOH	0.462	−20.562
	D28	CO-OCH_3_	0.477	−23.965
	D29	CH_3_	0.513	−26.233
	D30	CH_2_CH_3_	0.544	−28.266
	D31	CH_2_CH_2_CH_3_	0.605	−32.148
	D32	CH(CH_3_)_2_	0.573	−30.184
	D33	(CH_2_)_3_CH_3_	0.615	−32.682
	D34	CH(CH_3_)CH_2_CH_3_	0.631	−33.048
	D35	CH_2_CH(CH_3_)CH_3_	0.639	−33.865
	D36	Cl	0.399	−16.705
	D37	CH_2_Cl	0.505	−26.158
	D38	SH	0.428	−18.175
	D39	S-CH_3_	0.578	−30.349
	D40	NH_2_	0.473	−21.872

**Table 4 plants-13-01710-t004:** Structures and QED of new patulin’s derivatives, which have different group at 5-position, and the binding affinity between each compound and D1 protein of *A. thaliana* (5MDX). Five derivatives (D41, D43, D44, D45, and D46) show higher QED and binding affinity than patulin.

Chemical Structure	Compound No.	Substituent (R-)	QED	Interaction Energy (kal·mol^−1^)
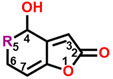	D0 (patulin)	O	0.488	−25.383
	D41	S	0.536	−27.948
	D42	NH	0.458	−19.896
	D43	N-CH_3_	0.515	−26.719
	D44	CH_2_	0.513	−26.264
	D45	CHCH_3_	0.540	−28.108
	D46	CHCH_2_CH_3_	0.617	−32.846
	D47	CHCl	0.447	−19.043
	D48	CH-OH	0.478	−24.125
	D49	CH-NH_2_	0.463	−20.694

**Table 5 plants-13-01710-t005:** Structures and QED of new patulin’s derivatives, which have different R group at the C6 position, and the binding affinity between each compound and *Arabidopsis* D1 protein (5MDX). Eleven derivatives (D50, D51, D52, D53, D54, D56, D57, D59, D65, D66, and D67) show higher QED and binding affinity than patulin.

Chemical Structure	Compound No.	Substituent (R-)	QED	Interaction Energy (kal·mol^−1^)
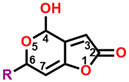	D0 (patulin)	H	0.488	−25.383
	D50	CH_3_	0.524	−27.559
	D51	CH_2_CH_3_	0.601	−30.941
	D52	CH_2_CH_2_CH_3_	0.671	−35.015
	D53	CH(CH_3_)_2_	0.634	−33.339
	D54	CH_2_Cl	0.494	−25.943
	D55	OH	0.474	−22.563
	D56	O-CH_3_	0.571	−30.058
	D57	O-CH_2_CH_3_	0.639	−34.215
	D58	CHO	0.431	−18.449
	D59	CO-CH_3_	0.569	−29.946
	D60	COOH	0.443	−18.982
	D61	CO-OCH_3_	0.425	−18.076
	D62	CO-OCH(CH_3_)_2_	0.414	−17.576
	D63	CO-OCH_2_CH_3_	0.458	−19.648
	D64	SH	0.420	−17.773
	D65	S-CH_3_	0.522	−26.93
	D66	CH_2_-S-CH_3_	0.713	−36.428
	D67	(CH_2_)_2_-S-CH_3_	0.671	−35.601
	D68	NH_2_	0.459	−20.083

**Table 6 plants-13-01710-t006:** Structures and QED of new patulin’s derivatives, which have different R group at the C7 position, and the binding affinity between each compound and *Arabidopsis* D1 protein (5MDX). Seven derivatives (D69, D70, D71, D73, D75, D77, and D79) show higher QED and binding affinity than patulin.

Chemical Structure	Compound No.	Substituent (R-)	QED	Interaction Energy (kal·mol^−1^)
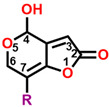	D0 (patulin)	H	0.488	−25.383
	D69	CH_3_	0.523	−27.408
	D70	CH_2_CH_3_	0.604	−31.806
	D71	CH_2_Cl	0.490	−25.368
	D72	OH	0.464	−21.156
	D73	O-CH_3_	0.552	−28.483
	D74	CHO	0.430	−18.435
	D75	CO-CH_3_	0.567	−29.442
	D76	COOH	0.442	−18.801
	D77	CO-OCH3	0.523	−27.143
	D78	SH	0.403	−17.282
	D79	S-CH_3_	0.601	−31.521
	D80	NH_2_	0.432	−18.358
	D81	Cl	0.481	−24.548

**Table 7 plants-13-01710-t007:** Toxicity risks of patulin and its derivatives, which show higher QED and lower interaction energy, were predicted by Discovery Studio. Rat oral lethal dose (LD_50_), rat inhalational LD_50_, skin irritancy, skin sensitization, ocular irritancy, and Ames mutagenicity were evaluated. The LD_50_ experimental data (exp.) of rat oral LD_50_ and Ames mutagenicity are also exhibited according to previous reports to validate the reliability of the predicted toxicity in this study.

Compound No.	Rat Oral LD_50_ (mg/kg)	Rat Inhalational LD_50_ (mg/m^3^)	Skin Irritancy	Skin Sensitization	Ocular Irritancy	Ames Mutagenicity
D0 (patulin)	55 (exp.)	-	-	-	-	Non-Mutagen (exp.)
	139.305	4338.681	Moderate	Moderate	Severe	Non-Mutagen
D1	571.536	3601.752	Mild	Weak	Severe	Non-Mutagen
D3	993.933	10,833.50	Mild	None	Mild	Non-Mutagen
D4	607.153	4613.415	Mild	Weak	Severe	Non-Mutagen
D5	369.025	9291.594	Mild	Weak	Severe	Non-Mutagen
D6	587.353	21,647.80	Mild	Weak	Mild	Non-Mutagen
D7	281.027	3239.335	Mild	Weak	Severe	Non-Mutagen
D8	591.034	2774.117	Mild	Weak	Severe	Non-Mutagen
D10	148.169	9713.14	Mild	Weak	Severe	Non-Mutagen
D12	168.457	3494.71	Mild	Weak	Severe	Non-Mutagen
D16	327.657	5498.03	Mild	Weak	Moderate	Non-Mutagen
D18	414.29	8957.64	Mild	None	Mild	Non-Mutagen
D19	61.1887	3218.41	Mild	Weak	Moderate	Non-Mutagen
D26	156.119	2402.14	Mild	Weak	Severe	Non-Mutagen
D29	157.887	12,885.25	Mild	Weak	Severe	Non-Mutagen
D30	368.658	30,435.47	Mild	Weak	Severe	Non-Mutagen
D31	410.105	55,470.60	Severe	None	Severe	Non-Mutagen
D32	306.655	32,070.50	Moderate	Weak	Severe	Non-Mutagen
D33	422.363	54,391.60	Moderate	None	Severe	Non-Mutagen
D34	500.764	74,265.90	Mild	Weak	Mild	Non-Mutagen
D35	418.021	33,862.10	Moderate	Weak	Mild	Non-Mutagen
D37	103.857	3289.45	Mild	Weak	Severe	Non-Mutagen
D39	232.301	5588.03	Mild	Weak	Mild	Non-Mutagen
D41	174.595	3667.57	Mild	Weak	Severe	Non-Mutagen
D43	293.806	11,837.50	Mild	Weak	Severe	Non-Mutagen
D44	198.445	4834.81	Mild	Weak	Severe	Non-Mutagen
D45	223.607	9603.58	Mild	Weak	Severe	Non-Mutagen
D46	213.791	22,374.60	Mild	Weak	Severe	Non-Mutagen
D50	235.67	6575.84	Mild	Weak	Severe	Non-Mutagen
D51	358.416	15,407.70	Mild	Weak	Severe	Non-Mutagen
D52	411.583	27,890	Mild	Weak	Severe	Non-Mutagen
D53	343.009	16,124.70	Mild	Weak	Severe	Non-Mutagen
D54	78.2202	1649.27	Mild	Weak	Severe	Non-Mutagen
D56	285.465	5630.42	Mild	Weak	Moderate	Non-Mutagen
D57	311.297	10,151	Mild	Weak	Moderate	Non-Mutagen
D59	266.431	5320.76	Mild	Weak	Moderate	Non-Mutagen
D65	270.03	2804.60	Mild	Weak	Severe	Non-Mutagen
D66	315.52	2784.97	Mild	Weak	Severe	Non-Mutagen
D67	457.744	2395.11	Mild	Weak	Mild	Non-Mutagen
D69	228.207	9713.14	Mild	Weak	Severe	Non-Mutagen
D70	159.021	20,286.60	Mild	Weak	Severe	Non-Mutagen
D71	59.1671	2143.11	Mild	Weak	Severe	Non-Mutagen
D73	124.718	7143.39	Mild	Weak	Severe	Non-Mutagen
D75	122.299	6986.65	Mild	Weak	Severe	Non-Mutagen
D77	409.641	5498.03	Mild	Weak	Moderate	Non-Mutagen
D79	142.33	3494.71	Mild	Weak	Severe	Non-Mutagen

**Table 8 plants-13-01710-t008:** Structures, quantitative estimate of drug-likeness (QED), possible bonding interactions, and the affinity of D0 (patulin), D3, D6, D34, and D67 binding to *Arabidopsis* D1 protein (5MDX).

Com.	Mol. Formula	Chemical Structure	QED	Bonding Donors	Bonding Acceptors	Interactions	Bound Distance (Å)	Interaction Energy (kal·mol^−1^)
D0 (patulin)	C_7_H_6_O_4_	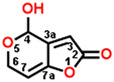	0.488	**D0 O2** D1-Phe255 D1-Ser264 D1-Phe265	**D1-His252 NH** D0 C4 D0 C3 D0 C6	**Hydrogen Bond** Pi-alkyl Hydrophobic Alkyl Hydrophobic Alkyl Hydrophobic	**2.38** 3.83 4.22 3.51	−25.383
D3	C_8_H_9_NO_3_	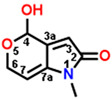	0.518	**D3 O2** D1-Phe255 D1-Ser264 D1-Phe265	**D1-His252 NH** D3 C4 D3 C3 D3 C6	**Hydrogen Bond** Alkyl Hydrophobic Pi-alkyl Hydrophobic Alkyl Hydrophobic	**2.27** 3.17 3.42 3.57	−26.871
D6	C_10_H_12_O_3_	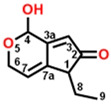	0.651	**D6 O2** D1-Phe255 D1-Ser264 D1-Leu271	**D1-His252 NH** D6 C3 D6 C9 D6 C6	**Hydrogen Bond** Pi-alkyl Hydrophobic Alkyl Hydrophobic Alkyl Hydrophobic	**2.13** 3.52 3.39 4.25	−34.373
D34	C_11_H_14_O_3_	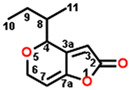	0.631	**D34 O2** D1-Phe255 D1-Phe260 D1-Ala263 D1-Leu271	**D1-His252 NH** D34 C11 D34 C10 D34 C10 D34 C11	**Hydrogen Bond** Pi-alkyl Hydrophobic Pi-alkyl Hydrophobic Alkyl Hydrophobic Alkyl Hydrophobic	**2.18** 3.21 3.30 3.44 3.58	−33.048
D67	C_10_H_12_O_4_S	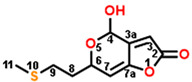	0.671	D1-His215 **D67 O2** D1-Leu271 D1-His272 D1-Leu275	D67 C8 **D1**-**His252 NH** D67 C9 D67 C11 D67 C11	Alkyl Hydrophobic **Hydrogen Bond** Alkyl Hydrophobic Alkyl Hydrophobic Alkyl Hydrophobic	3.09 **2.06** 3.15 3.42 3.33	−35.601

## Data Availability

All data presented in this study are contained in the main text.

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
