# Peer review of "Structure-Based Design, Virtual Screening, and Discovery of Novel Patulin Derivatives as Biogenic Photosystem II Inhibiting Herbicides"

_plants, 2024, doi:10.3390/plants13121710_

Round 1

Reviewer 1 Report

Comments and Suggestions for Authors

In the present manuscript authors describe the investigation of the “Structure-based design, virtual screening and discovery of novel patulin derivatives as biogenic PSII inhibiting herbicides” Quite a lot of experiments were involved in these studies; thus, the work is solid. The introduction is clear and shows the literature data about the man subject. Methodology is adequate. Results seem fine and appear correct.

The manuscript can be accepted after some revision, as shown below.

 The authors said that the unsaturated lactone containing C=O group at 2-position is the core part that bounds to the QB site by forming hydrogen bonds to histidine in the D1 protein.

They found that the substituents (R) including alkyl side chain and methylthio group introduced to the patulin of ether positions (neither position of the following compounds would be an ether; so, I suggest writing to the patulin at positions indicated next to each code), can increase the quantitative estimate of drug-likeness (QED) value, e.g. compounds D4-D6 (1-), D10 (C-3), D12 (C-3), D29-D35 (C-4), D39 (C-4), D44-D46 (5-), D50-D53 (C-6), D65-D67 (C-6), D69 (C-7), D70  (C-7) and D79  (C-7). Methylamino and acylamide groups introduced to the patulin at 1-, 2- (correct 4-) or 5-position, such as compounds D3, D18 and D43, can increase the QED value. The QED value of the compounds will be increased while the sulfur is introduced in the patulin at 1- or 5-position, or chlorine group (this is wrong, as it means chlorine alone, the correct one would be alkyl chloride) is introduced in the patulin at 3-, C4 (CH2Cl) or C6 (CH2Cl) position. The introduction of ether group at the C6 or C7 position can also increase the QED value. The QED values of compounds increase when the ester group is introduced at the C3 or C7 position. However, introducing of hydrophilic groups involving hydroxy, aldehyde, carboxy, chloride (Cl), sulfhydryl or amino leads to a decrease in the QED value and the binding affinity.

I suggest that the authors comment on the resonance and inductive electronic effects of the respective substituents analyzed. The authors said that the unsaturated lactone containing C=O group at 2-position is the core part that bounds to the QB site by forming hydrogen bonds to histidine in the D1 protein. Thus, I would expect that groups in the 1- and C-7 position, which donate electrons by resonance to the C=O bond, or to the double bonds at C-7 up to the carboxyl bond would reinforce hydrogen bond formation with histidine, and thus vary QED. These discussions will make the results more scientifically reliable.

 Table 8: doubts: - D0 C8, where this position is in the structure.

- Pi-alkyl Hydrophobic: what does Pi mean, if it is a π bond referring to the carbon-carbon double bond, there is no π bond in these models.

 Figure 2: Patulin structure in frame A is wrong, review.

Comments on the Quality of English Language

Only minor editing of English language is required.

Author Response

Thank you very much for taking the time to review this manuscript. Please find the detailed responses in the attachment, and the corresponding revisions/corrections highlighted/in track changes in the re-submitted files.

Reviewer 2 Report

Comments and Suggestions for Authors

The manuscript “Structure-based design, virtual screening and discovery of novel patulin derivatives as biogenic PSII inhibiting herbicides” is dealing with computer-aided design of patulin derivatives and prediction of their possible toxicity and herbicide potency. Patulin is mycotoxin with significant herbicidal activity to various weeds. Authors reported eighty-one newly designed derivatives, among them forty-five derivatives predicted better affinities than patulin and were chosen for further toxicity evaluation. According to computer analysis, four novel compounds with substituents of the methylamino (D3) or propyl (D6) at 1-position, sec-butyl at 4-position (D34) or methylthio at the 9-position (D67) showed higher affinity to the QB site than patulin, lower toxicity than patulin and were predicted to have a great potential to develop as new herbicides with improved potency. However, further studies are required to synthesize predicted compounds. Also it is necessary to evaluate their PSII inhibitory capacity, herbicidal activity, and animal toxicology for possible commercial development.

The results presented in manuscript has some theoretical merit and it is acceptable for publication in journal Plants. According to aim and the scope the journal it published also theoretical results of research in all fundamental and applied fields of plant science.

There are some minor errors which are highlighted in text:

Line 108: Reference is missing

Lines 263-280: Why all this section is in bold?

Line 280 Material and methods as separate subheading

Line 359 Five or four?

Author Response

Thank you very much for taking the time to review this manuscript. Please find the detailed responses in the attachment and the corresponding revisions/corrections highlighted/in track changes in the re-submitted files.

Reviewer 3 Report

Comments and Suggestions for Authors

The introduction presents sufficient data supported by  references and clearly presents the purpose of the research. The method is clearly presented, the simulated models of patulin derivatives are presented and an attempt was made to theoretically identify the derivatives which show higher quantitative estimate of drug-likeness (QED) and binding affinity than patulin. Based on data from various researches, 4 derivatives with bioherbicide potential are identified, but a more detailed characterization of them would be recommended, like their degradation. The degradation of an herbicide results in several compounds ("metabolites"), which have their own chemical properties, including toxicity, adsorption capacity and resistance to degradation. Some metabolites are more toxic and/or persistent than the parent compound

Author Response

(The authors gave the same response as above.)

Round 2

Reviewer 1 Report

Comments and Suggestions for Authors

The authors responded to all the suggestions made, as well as correcting some errors indicated previously. Therefore, I inform you that the work can be accepted for publication.